# MixFeat: Mix Feature in Latent Space Learn Discriminative Space

## Abstract

Deep learning methods perform well in various tasks. However, the over-fitting problem, which causes the performance to decrease for unknown data, remains. We hence propose a method named *MixFeat* that directly creates latent spaces in a network that can distinguish classes. MixFeat mixes two feature maps in each latent space in the network and uses unmixed labels for learning. We discuss the difference between a method that mixes only features (MixFeat) and a method that mixes both features and labels (mixup and its family). Mixing features repeatedly is effective in expanding feature diversity, but mixing labels repeatedly makes learning difficult. MixFeat makes it possible to obtain the advantages of repeated mixing by mixing only features. We report improved results obtained using existing network models with MixFeat on CIFAR-10/100 datasets. In addition, we show that MixFeat effectively reduces the over-fitting problem even when the training dataset is small or contains errors. MixFeat is easy to implement and can be added to various network models without additional computational cost in the inference phase.

## 1 Introduction

Deep neural networks (LeCun et al., 1998) have performed well for various tasks, such as image recognition (Krizhevsky et al., 2012; Simonyan & Zisserman, 2015; He et al., 2016a;b; Han et al., 2017; Huang et al., 2017), object detection (Ren et al., 2015; Redmon et al., 2016), and semantic segmentation (Chen et al., 2018; Badrinarayanan et al., 2017).

One remaining problem with training deep neural networks is the over-fitting of training data, despite many methods having been proposed to solve this problem; e.g., the dropout method (Srivastava et al., 2014) drops randomly selected elements of feature maps, the mixup method (Zhang et al., 2018a) and between-class learning (Tokozume et al., 2018) mix pairs of training images and labels, and the manifold mixup method (Verma et al., 2018) mixes pairs of training feature maps on a randomly selected latent space and labels. We propose a method named MixFeat to make each latent space in the network better distinguish each class.

The main contributions of this paper are as follows.

- We propose the MixFeat scheme to reduce the over-fitting problem by mixing two feature maps in each latent space in the network so that it can distinguish each class without any additional computational cost in the inference phase.

- We conduct extensive experiments to demonstrate the effectiveness of the generalization of MixFeat.

MixFeat is easy to implement and can be added to various neural network models. It has the potential to be applied for various tasks, such as object detection, semantic segmentation, and anomaly detection. MixFeat is described in the next section.

## 2 MIXFEAT

### 2.1 OVERVIEW

To avoid overfitting, we consider a method that creates latent spaces in a neural network such that each space can distinguish all classes. Training a network fed with a perturbed sample to output the same inference as when it is fed the pure sample enlarges Fisher's criterion (Fisher, 1936) (*i.e.*, the ratio of the between-class distance to the within-class variance). It is therefore conceivable for a network to learn the perturbed samples in order to make each latent space able to distinguish the classes. However, a perturbation that is independent of the given examples is inefficient because the latent space is extremely high-dimensional and dynamically changes during learning. We consider that the perturbation should be determined according to the subspace spanned by several samples in the latent space. For simplification, we adopt a partial plane spanned by the origin, the base example, and another example, as shown in Fig 6(a). In this study, we propose the MixFeat method, which mixes two feature maps in the latent spaces in the network to create latent spaces that can distinguish each class.

### 2.2 MIXING ONLY FEATURES OR BOTH FEATURES AND LABELS

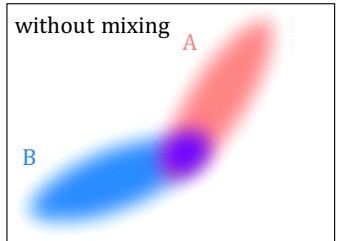 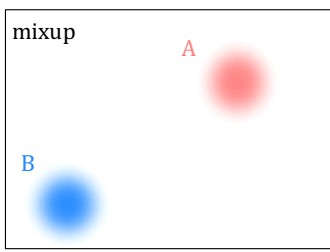 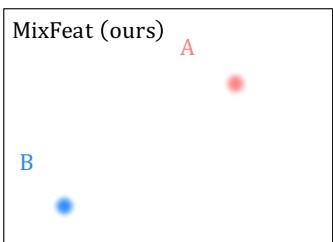

Figure 1:   Class distributions in the learned latent spaces, where A and B respectively denote the distributions of classes A and B while $A_i$ and $B_j$ respectively denote examples in classes A and B. The mixed features are distributed such that it is possible to distinguish each class in the mixup method and the mixed features are distributed such that it is easier to distinguish each class in MixFeat. Because MixFeat repeatedly mixes training examples in each latent space to extend diversity, the obtained feature space is able to distinguish each class more easily.

In related studies, methods that mix two images and labels have recently been proposed. The mixup method (Zhang et al., 2018a), between-class learning (BCL) (Tokozume et al., 2018), and manifold mixup method (Verma et al., 2018) mix two training examples both with respect to features and labels with random weights, and they have been reported to appreciably reduce over-fitting.

We argue that mixing methods are able to constrain the feature distribution, which cannot be achieved by training without mixing, and the feature space obtained by MixFeat differs greatly from those obtained by the mixup family of methods. Figure 1 shows the assumed class distributions in learned feature spaces. Without mixing, the feature distribution of the mixed features increases and overlaps the feature distributions of classes A and B (Fig. 1(left)). Mixing methods reduce the feature distribution of each class and make it easy to distinguish between classes using the following schemes:

- Regression learning: a learning method that associates mixed features with mixed labels;
- Learning against perturbation: a learning method that associates mixed features with unmixed labels.

Moreover, we consider repeatedly mixing features in various layers to obtain more diverse features. Learning highly diverse features makes it easier for a latent space to distinguish each class and better suppress over-fitting problems. However, learning to associate repeatedly mixed features with repeatedly mixed labels is intuitively difficult. This behavior was also reported in Tokozume et al. (2018). The authors stated that "*the performance when we used only the mixtures of three different classes ($N = 3$) was worse than that of $N = 2$ despite the larger variation in training data*," which

indicates that we cannot mix repeatedly in the mixup family of methods. In contrast, learning repeatedly perturbed features is easy because the perturbation parameter is not a learning target. Hence, we adopt a method that mixes only features so that repeatedly mixed features can be effectively learned. MixFeat repeatedly mixes features in one training pass, so MixFeat achieves a latent space that can distinguish each class more easily (Fig. 1(right)).

## 2.3 MixFeat Algorithm

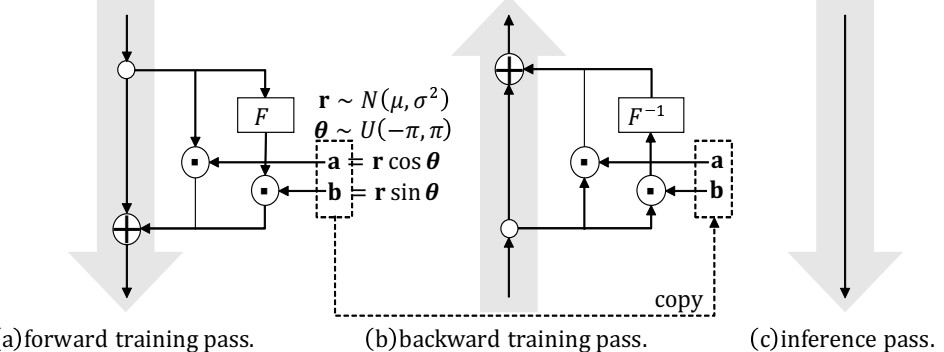

(a)forward training pass.  (b)backward training pass.  (c)inference pass.

Figure 2: Computational passes of MixFeat. Let $\oplus$ denote the addition operator and $\odot$ denote the sample-wise product. Vector $\mathbf{r}$ is a random-value vector sampled from Gaussian distribution $N(0, \sigma^2)$, whereas vector $\boldsymbol{\theta}$ is a random-value vector sampled from uniform distribution $U(-\pi, \pi)$, where $\pi$ indicates the circular constant. Each element of $\mathbf{r}$ or $\boldsymbol{\theta}$ is associated with the feature map of each sample in the mini-batch. Furthermore, $F$ denotes the random sort operation along the example axis to the input tensor, $F^{-1}$ denotes the restoring order operation along the example axis to the input tensor, and (*copy*) denotes copying the vector from the forward training pass. The inference phase returns the input tensor without modification.

The process of MixFeat is shown in Fig. 2. Let $\oplus$ denote the addition operator and $\odot$ denote the sample-wise product.

The forward training pass, as shown in Fig. 2(a), is described as

$$\boldsymbol{Y} = \boldsymbol{X} + (\mathbf{a} \odot \boldsymbol{X} + \mathbf{b} \odot F(\boldsymbol{X})), \tag{1}$$

$$\text{where } \mathbf{a} = \mathbf{r} \cos \boldsymbol{\theta}, \ \mathbf{b} = \mathbf{r} \sin \boldsymbol{\theta}, \ \mathbf{r} \sim N(0, \sigma^2), \ \boldsymbol{\theta} \sim U(-\pi, \pi),$$

where the first term $\boldsymbol{X}$ is the input mini-batch tensor, the second term $\mathbf{a} \odot \boldsymbol{X} + \mathbf{b} \odot F(\boldsymbol{X})$ is the perturbation, $\boldsymbol{Y}$ denotes the output mini-batch tensors of MixFeat, $\mathbf{r}$ is a Gaussian random vector with $N(0, \sigma^2)$, $\boldsymbol{\theta}$ is a uniform random vector with $U(-\pi, \pi)$, and $\pi$ is the circular constant. Each element of $\mathbf{r}$ or $\boldsymbol{\theta}$ is associated with the feature map of each sample in the mini-batch. Furthermore, $F(\boldsymbol{X})$ denotes $\boldsymbol{X}$ randomly sorted along the example axis. An appropriate value for $\sigma$ is discussed later.

The backward training pass, shown in Fig.2(b), is calculated as

$$\boldsymbol{G_X} = \boldsymbol{G_Y} + (\mathbf{a} \odot \boldsymbol{G_Y} + F^{-1}(\mathbf{b} \odot \boldsymbol{G_Y})), \tag{2}$$

where $\boldsymbol{G_X}$ and $\boldsymbol{G_Y}$ respectively denote the partial derivatives of the final output loss function with respect to $\boldsymbol{X}$ and $\boldsymbol{Y}$, $F^{-1}(\cdot)$ denotes the inverse operation of $F(\cdot)$, which restores the order of examples before the random sorting, and vectors $\mathbf{a}$ and $\mathbf{b}$ are copies of $\mathbf{a}$ and $\mathbf{b}$ in the forward training pass (*copy* in Fig. 2).

During inference, as shown in Fig. 2(c), the perturbation branches are not necessary for inference and can be removed as follows:

$$\boldsymbol{Y} = \boldsymbol{X}. \tag{3}$$

Equation (3) indicates that the MixFeat layer in the inference phase returns the input as is; i.e., the MixFeat layer can be simply removed from the inference phase and it thus does not lead to any additional computational cost at this point.

## 3 EXPERIMENTS

### 3.1 CIFAR-10 AND 100 DATASETS

The following experiments were conducted on the CIFAR-10 and -100 datasets (Krizhevsky & Hinton, 2009). The two CIFAR datasets consist of RGB natural images comprising $32 \times 32$ pixels. CIFAR-10 consists of images drawn from 10 classes while CIFAR-100 is drawn from 100 classes. The CIFAR-10 and -100 datasets respectively contain 50,000 training images and 10,000 test images. In our experiments, the input images of the CIFAR-10 and -100 datasets were processed adopting the following conventional augmentation process (Krizhevsky et al., 2012; Simonyan & Zisserman, 2015). The original image of $32 \times 32$ pixels was color-normalized and then horizontally flipped with 50% probability. It was then zero-padded to a size of $40 \times 40$ pixels and randomly cropped to an image of $32 \times 32$ pixels.

All models were trained employing back-propagation and a stochastic gradient descent with Nesterov momentum (Sutskever et al., 2013). We adopted the weight initialization introduced by He et al. (2015). A single graphics processing unit (GeForce GTX Titan X or GeForce GTX 1080 Ti) was used for each training. The initial learning rate was set to 0.05 and decayed by a factor of 0.1 at the half and three-quarter points of the overall training process (300 epochs), following Huang et al. (2017). In addition, we used a weight decay (Krogh & Hertz, 1992) of $5 \times 10^{-4}$, momentum of 0.9, and batch size of 128.

When using the proposed MixFeat method, MixFeat is placed directly after each convolution and $\sigma = 0.2$ is used unless otherwise specified.

We compared the performance of the proposed MixFeat method with two other mixing methods: the mixup method (Zhang et al., 2018a) and manifold mixup method (Verma et al., 2018). We trained a ResNet (pre-activation version) (He et al., 2016b), DenseNet, DenseNetBC, (Huang et al., 2017), and PyramidNet (Han et al., 2017). Following the settings of the original papers, the distribution parameter $\alpha = 1.0$ was used for the mixup method and $\alpha = 2.0$ was used for the manifold mixup method. For PyramidNet, the initial learning rate was set to 0.01 and a batch size of 32 was used depending on the memory limitations of the graphics processing unit. We implemented the methods using Chainer v4.4.0 (Tokui et al., 2015).

Results are given in Table 1. The results obtained with the mixup method were consistently better than those obtained with the vanilla model, which is a model that does not avoid over-fitting. In addition, the manifold mixup method sometimes achieved better result than original mixup method. Ultimately, the best performance was obtained when MixFeat was adopted for almost all network models. The best results on the CIFAR-10 and -100 datasets were 2.92% and 16.03% for the 272-layer PyramidNet. These results demonstrate that MixFeat improves the performance of various network models.

The training and test error curves obtained with and without MixFeat are shown in Fig. 3. The test error rate training with MixFeat shows better convergence than the rate training without MixFeat. In addition, the training and testing error curves for training with MixFeat are closer together than those for training without MixFeat, which demonstrates that MixFeat reduces over-fitting.

### 3.2 VARIOUS REGULARIZATION METHODS

We compared the performance of the proposed MixFeat method with several other over-fitting avoidance methods: the mixing methods mixup (Zhang et al., 2018a), manifold mixup (Verma et al., 2018), and input MixFeat; and the shake methods Shake-shake (Gastaldi, 2017), ShakeDrop (Yamada et al., 2018), and swapout (Singh et al., 2016). Input MixFeat indicates a method that mixes only input images (and does not mix labels) as a special case of MixFeat. We trained ResNet (the pre-activation version) (He et al., 2016b) with the learning settings reported in 3.1. Additionally, following the original papers, stochastic parameters $\theta_1 = \theta_2 = \text{Linear}(1, 0.5)$ were used for swapout and $b_l = \text{Linear}(1, 0.5)$ was used for ShakeDrop; shake parameters $\alpha_i \sim U(0, 1), \beta_i \sim U(0, 1)$ were used for the shake-shake and $\alpha \sim U(-1, 1), \beta \sim U(0, 1)$ were used for ShakeDrop.

Table 1: Benchmark results on the CIFAR-10 and -100 datasets. We show the average and standard errors for five trials. C10 and C100 respectively indicate CIFAR-10 and CIFAR-100. ResNet-B indicates ResNet with bottleneck module. For hyperparameters, we used $k = 12$ in DenseNet/DenseNet-BC and $\alpha = 200$ in PyramidNet. The order of the layers in a module are the same in each model except of the PyramidNet: BN-ReLU-Conv (ResNet/DenseNetBC repeat this order twice for each module and ResNet-B repeat this order three times for each module). The order of the layers in a module in PyramidNet is: BN-Conv-BN-ReLU-Conv-BN-ReLU-Conv-BN. *Vanilla* indicates that nothing was done to avoid over-fitting, *mixup* (Zhang et al., 2018a) and *manifold mixup* (Verma et al., 2018) are used as the learning schemes, and *MixFeat* is located immediately after each convolution layer. Overall, MixFeat improves the performance of all convolutional neural network models.

| Dataset | Model | Depth | #Params | Test error rate (%) | | | |
| --- | --- | --- | --- | --- | --- | --- | --- |
| | | | | Vanilla | Mixup | Manifold mixup | MixFeat (ours) |
| C10 | ResNet | 20 | 0.3M | $7.33 \pm 0.09$ | $7.02 \pm 0.12$ | $7.68 \pm 0.14$ | $\mathbf{6.54 \pm 0.24}$ |
| | ResNet | 110 | 1.7M | $5.18 \pm 0.10$ | $\mathbf{4.48 \pm 0.15}$ | $4.95 \pm 0.08$ | $4.78 \pm 0.18$ |
| | ResNet-B | 164 | 1.7M | $4.61 \pm 0.07$ | $3.86 \pm 0.11$ | $3.96 \pm 0.07$ | $\mathbf{3.78 \pm 0.04}$ |
| | DenseNet | 40 | 1.0M | $5.59 \pm 0.20$ | $5.12 \pm 0.12$ | $5.43 \pm 0.16$ | $\mathbf{4.82 \pm 0.10}$ |
| | DenseNet-BC | 100 | 0.8M | $4.66 \pm 0.12$ | $4.17 \pm 0.13$ | $4.67 \pm 0.19$ | $\mathbf{3.91 \pm 0.13}$ |
| | PyramidNet | 272 | 26.0M | $3.64 \pm 0.15$ | - | - | $\mathbf{2.92 \pm 0.06}$ |
| C100 | ResNet | 20 | 0.3M | $31.39 \pm 0.26$ | $30.35 \pm 0.40$ | $31.33 \pm 0.39$ | $\mathbf{29.67 \pm 0.23}$ |
| | ResNet | 110 | 1.7M | $24.59 \pm 0.13$ | $22.63 \pm 0.41$ | $\mathbf{22.58 \pm 0.23}$ | $23.58 \pm 0.46$ |
| | ResNet-B | 164 | 1.7M | $21.31 \pm 0.23$ | $19.50 \pm 0.22$ | $\mathbf{19.18 \pm 0.27}$ | $19.92 \pm 0.24$ |
| | DenseNet | 40 | 1.0M | $26.40 \pm 0.23$ | $\mathbf{23.31 \pm 0.19}$ | $24.17 \pm 0.34$ | $\mathbf{23.31 \pm 0.28}$ |
| | DenseNet-BC | 100 | 0.8M | $22.98 \pm 0.13$ | $20.80 \pm 0.22$ | $20.85 \pm 0.24$ | $\mathbf{20.09 \pm 0.16}$ |
| | PyramidNet | 272 | 26.0M | $17.74 \pm 0.23$ | - | - | $\mathbf{16.03 \pm 0.15}$ |

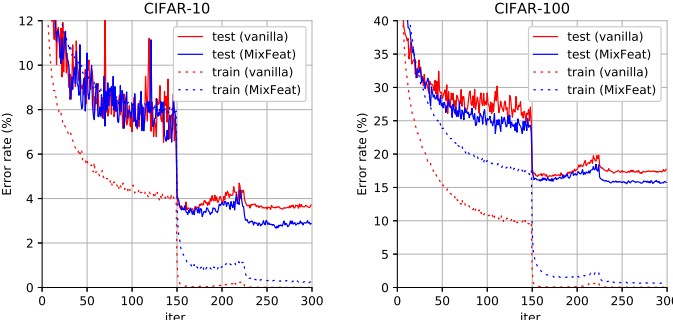

Figure 3: Training and test error curves for PyramidNet ($depth = 272, \alpha = 200$). **Left:** Training curve on CIFAR-10. **Right:** Training curve on CIFAR-100. The discrepancy between the training and test curves is suppressed by MixFeat.

Results are given in Table 2. The results obtained by the mixing methods are better than those obtained by the non-mixing methods, and MixFeat achieves the best performances. Comparing the MixFeat methods, the method that mixes features in each latent feature space is much better than the method that only mixes features at the input stage, both in CIFAR-10 and CIFAR-100. The shake methods seem to insufficiently converge even with 300 epochs of training, as mentioned in their original papers (Gastaldi, 2017; Yamada et al., 2018).

### 3.3 OVER-FITTING AVOIDANCE PERFORMANCE

We present experimental results that demonstrate how well MixFeat avoids over-fitting. Additionally, we experimentally confirm that MixFeat and mixup are different approaches to reduce over-fitting.

Table 2: Comparison of the results of various regularization methods on the CIFAR-10 and -100 datasets using a 20-layer ResNet (pre-activation). *Vanilla* indicates that nothing was done to avoid over-fitting, *mixup* (Zhang et al., 2018a) and *manifold mixup* (Verma et al., 2018) are used as the learning schemes, and *MixFeat* is located immediately after each convolution layer. *Shake-shake* (Gastaldi, 2017), *ShakeDrop* (Yamada et al., 2018), and *swapout* (Singh et al., 2016) are shake methods. Moreover, shake-shake has parallel convolution layers and hence twice the number of parameters of the other methods. *Input MixFeat* mixes only input images (not labels) as a special case of MixFeat. Overall, MixFeat achieves the best performance.

| | 20-layer ResNets | |
| | Test error rate (%) | |
| Method | CIFAR-10 | CIFAR-100 |
| --- | --- | --- |
| Vanilla | $7.33 \pm 0.09$ | $31.39 \pm 0.26$ |
| Mixup | $7.02 \pm 0.12$ | $30.35 \pm 0.40$ |
| Manifold mixup | $7.68 \pm 0.14$ | $31.33 \pm 0.39$ |
| MixFeat (ours) | $\mathbf{6.54 \pm 0.24}$ | $\mathbf{29.67 \pm 0.23}$ |
| Shake-shake | $13.87 \pm 0.44$ | $51.31 \pm 1.64$ |
| ShakeDrop | $7.32 \pm 0.15$ | $31.49 \pm 0.30$ |
| Swapout | $8.17 \pm 0.24$ | $33.51 \pm 0.40$ |
| Input MixFeat | $6.96 \pm 0.18$ | $30.81 \pm 0.05$ |

### 3.3.1 WITH INCORRECT LABELS IN THE TRAINING DATASET

Incorrect labels in the training dataset degrade the test error rates because of over-fitting. We thus compared the test error rates with and without MixFeat while changing the ratio of incorrect labels in the training dataset. To construct a training dataset with a percentage $p$ of incorrect labels from the original CIFAR-10 training dataset, we randomly selected a percentage $p$ samples of training dataset and changed each label to another label that was randomly selected from the remaining nine classes. Results are shown in Fig. 4. As shown in Fig. 4 (Left), increasing the ratio of incorrect labels in the training data greatly magnifies the error rate without MixFeat whereas the increase in the error rate is considerably suppressed when MixFeat is used. As shown in Fig. 4 (Center and Right), the test curves degrade drastically after the peak without MixFeat whereas the test curves are kept low by MixFeat.

Mixup demonstrates the behavior same as MixFeat, and the method combined MixFeat and mixup has better performance than of each method, as shown in Fig. 4. The results confirm that MixFeat and mixup are different approaches and even compatible each other.

### 3.3.2 WITH SMALL TRAINING DATASETS

In general, learning with a small training dataset causes over-fitting. We thus compared the test error rates with and without MixFeat while reducing the size of the training dataset. In this experiment, the number of parameter update iterations was kept equal by increasing the number of epochs in inverse proportion to the training dataset size. Figure 5 (Left) shows that reducing the number of training data greatly increases the error rate without MixFeat whereas the increase in the error rate is considerably suppressed when MixFeat is used.

Mixup demonstrates the behavior same as MixFeat, and the method combined MixFeat and mixup has better performance than of each method, as shown in Fig. 5. The results confirm that MixFeat and mixup are different approaches and even compatible each other.

The results of these experiments indicate that MixFeat prevents deep neural networks from over-fitting caused by poor quality training data.

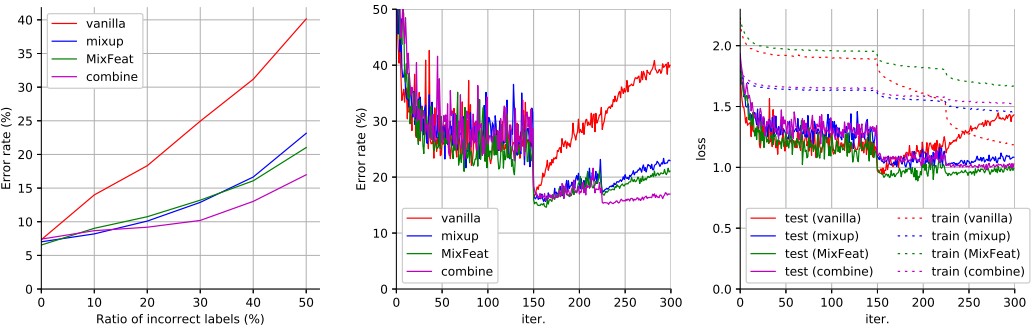

Figure 4: Results with incorrect labels in the training dataset without mixing, with MixFeat, with mixup, and with the method combined MixFeat and mixup on CIFAR-10 using 20-layer ResNets (pre-activation). **Left:** Test error (%) results for an increasing number of incorrect labels in the training dataset. The increase in the error rate as the number of incorrect labels increases is suppressed by MixFeat and mixup. Additionally, the method combined those two has better performance than of each method. **Center:** Test error curves for the training dataset with 50% incorrect labels. **Right:** Training and test loss curves for the training dataset with 50% incorrect labels. The test curves degrade drastically after the peak without mixing whereas they are kept low by MixFeat and mixup. Additionally, the method combined those two has better performance than of each method.

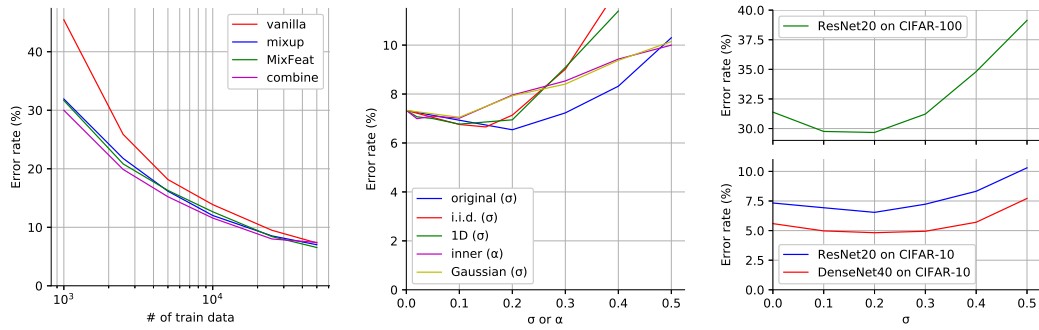

Figure 5: **Left:** Test error (%) results when reducing the training dataset size without mixing, with MixFeat, with mixup, and the method combined MixFeat and mixup on CIFAR-10 using 20-layer ResNets (pre-activation). The increase in the error rate with data reduction is suppressed with MixFeat and mixup. Additionally, the method combined those two has better performance than of each method. **Center:** Comparison of the results of original MixFeat, i.i.d.-MixFeat, 1D-MixFeat, inner-MixFeat, and Gaussian noise with changing hyperparameters $\sigma$ or $\alpha$ on CIFAR-10 using 20-layer ResNets (pre-activation). The original MixFeat has the highest performance. **Right:** Comparison of the results for various hyperparameter $\sigma$ values in MixFeat for various settings. The best value is $\sigma = 0.2$ in each setting.

## 3.4 ABLATION ANALYSIS

### 3.4.1 DIMENSIONS AND DIRECTION OF THE DISTRIBUTION

For perturbation factors $\mathbf{a}$ and $\mathbf{b}$ in Eq. 1, we can consider two types of distribution: one where the perturbation quantity follows Gaussian distribution and another where the spatial extent of the perturbation follows Gaussian distribution. MixFeat adopts the former concept, and here we compare to the latter concept as

$$\mathbf{a} \sim N(0, \sigma^2), \ \mathbf{b} \sim N(0, \sigma^2). \tag{4}$$

We call this variation i.i.d.-MixFeat. Intuitively, the difference between the two concepts is that the former perturbation distribution is concentrated near zero.

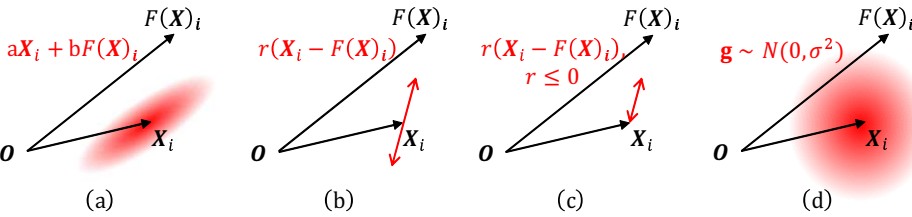

Figure 6: Distribution of various perturbations: (a) MixFeat / i.i.d.-MixFeat, (b) 1D-MixFeat, (c) inner-MixFeat, and (d) Gaussian noise.

Additionally, we can simply modify MixFeat as a one-dimensional version (1D-MixFeat) and inner-division version (inner-MixFeat), as shown in Fig. 6. The forward training pass of 1D-MixFeat is described as

$$\boldsymbol{Y} = \boldsymbol{X} + \mathbf{r} \odot (\boldsymbol{X} - F(\boldsymbol{X})), \tag{5}$$

whereas the backward training pass is described as

$$\boldsymbol{G_X} = \boldsymbol{G_Y} + (\mathbf{r} \odot \boldsymbol{G_Y} - F^{-1}(\mathbf{r} \odot \boldsymbol{G_Y})), \tag{6}$$

and the inference phase is the same as that of the original MixFeat.

Inner-MixFeat follows the mixing concept of the mixup method and BCL. The forward training pass is described as

$$\boldsymbol{Y} = \boldsymbol{X} + \mathbf{r}' \odot (\boldsymbol{X} - F(\boldsymbol{X})) \quad \text{where} \quad \mathbf{r}' = |B(\alpha, \alpha) - 0.5| - 0.5, \tag{7}$$

where $\mathbf{r}'$ is the mixing ratio based on a random vector with a beta distribution $B(\alpha, \alpha)$ following the mixup method. Note that we modified the mixing ratio to ensure consistency between the major component of the mixed image and label. The backward training pass is described as

$$\boldsymbol{G_X} = \boldsymbol{G_Y} + (\mathbf{r}' \odot \boldsymbol{G_Y} - F^{-1}(\mathbf{r}' \odot \boldsymbol{G_Y})) \tag{8}$$

and the inference phase is

$$\boldsymbol{Y} = (1 + \mathbb{E}\mathbf{r}')\boldsymbol{X}, \tag{9}$$

where $\mathbb{E}\mathbf{r}'$ denotes the expected value of $\mathbf{r}'$.

Finally, we also compared our MixFeat to a method that perturbs using isotropic Gaussian noise to demonstrate the superiority of directional perturbation. The forward training pass of the above perturbation (called Gaussian noise) is shown in Fig. 6 (d) and described as

$$\boldsymbol{Y} = \boldsymbol{X} + \mathbf{g} \tag{10}$$

$$\mathbf{g} \sim N(0, \sigma^2) \tag{11}$$

whereas the backward training pass is described as

$$\boldsymbol{G_X} = \boldsymbol{G_Y}, \tag{12}$$

and the inference phase is the same as that of MixFeat.

Figure 5 (Center) compares the test error rate with changing hyperparameter $\sigma$ or $\alpha$ for the original MixFeat, i.i.d.-MixFeat, 1D-MixFeat, inner-MixFeat, and Gaussian noise methods. The best results for each variation of perturbations were $6.54\%(\sigma = 0.2)$ for the original MixFeat, $6.66\%(\sigma = 0.15)$ for the i.i.d.-MixFeat, $6.77\%(\sigma = 0.1)$ for 1D-MixFeat, $6.94\%(\alpha = 0.02)$ for inner-MixFeat, and $7.05\%(\sigma = 0.1)$ for Gaussian noise. The original MixFeat thus has the highest performance.

### 3.4.2 LOCATION OF MIXFEAT IN THE NETWORK

We investigated the location of MixFeat in a commonly used pre-activation unit (He et al., 2016b; Huang et al., 2017), which consists of convolution (*Conv*), batch normalization (*BN*) (Ioffe & Szegedy, 2015), and a rectified linear unit (*ReLU*) (Nair & Hinton, 2010), referred to as a *-BN-ReLU-Conv-* network. Table 3 shows that MixFeat improve results regardless of the location, but the best location is after convolution. We therefore place MixFeat directly after each convolution.

### 3.4.3 REASONABLE HYPERPARAMETER VALUE $\sigma$

We investigated the standard deviation $\sigma$ of the distribution of $\mathbf{r}$ in Eq. (1). This investigation thus elucidates the optimal range of the distribution for effective perturbation. Figure 5 (Right) compares the results. Here, a value of $\sigma$ that is too small does not improve the performance while a value of $\sigma$ that is too large decreases the performance appreciably. That is to say, the majority of input tensor components of $\mathbf{Y}$ are replaced in the perturbation tensor if $|\mathbf{r}|$ is too large, and this range of values thus does not work well. Although $\sigma$ cannot be theoretically determined, $\sigma = 0.2$ is the experimentally determined optimal hyperparameter.

Table 3: Comparison of MixFeat locations in a *-(1)-BN-(2)-ReLU-(3)-Conv-(4)-* pre-activation unit. MixFeat is performed regardless of the location, but the best location is after convolution.

| 20-layer ResNet on CIFAR-10 | |
|---|---|
| MixFeat location | Error rate (%) |
| (no MixFeat) | 7.33 |
| (1) | 6.74 |
| (2) | 6.85 |
| (3) | 6.66 |
| (4) | **6.54** |

## 4 RELATIONSHIP WITH PREVIOUS WORK

We discuss the relationship between our approach and others that reduce the over-fitting of training data. Our approach is related to a series of approaches based on perturbing training data. We describe the differences between our approach and the others in the following.

Data augmentation methods for input images are widely used. The conventional data augmentation method (Krizhevsky et al., 2012; Simonyan & Zisserman, 2015) adds perturbations to the input images through geometric or value transformations. The cutout (DeVries & Taylor, 2017b) and random erasing (Zhong et al., 2017) methods overwrite elements in randomly selected rectangular regions with zeros or random values. These methods are intuitive and easily adjustable. Reasonable perturbation on the input image is independent of our method and a synergistic effect can be expected when using these methods together with our MixFeat method.

"Drop" perturbations are used for regularization and/or convergence acceleration. Dropout (Srivastava et al., 2014) drops randomly selected elements of feature maps, Dropconnect (Wan et al., 2013) drops randomly selected network connections, and ResDrop (Huang et al., 2016) drops randomly selected residual paths in ResNets (He et al., 2016a). It seems these methods have not been used much in recent times owing to their poor compatibility with batch normalization or their complicated implementations. However, these methods can be used with MixFeat if needed.

"Shake" methods calculate randomly weighted sums of parallel network branches. Shake-shake (Gastaldi, 2017) mixes the identity map and two residual branches with i.i.d. random weights for forward and backward passes. ShakeDrop (Yamada et al., 2018) mixes the identity map and one residual branch with independent (and not identically) random weights for forward and backward pass. They acquire an ensemble effect in each "shake" block. However, these methods worsen convergence speeds, as reported in Gastaldi (2017) and Yamada et al. (2018). These studies reported that 1,800-epoch training was better on CIFAR datasets, compared with 300-epoch training, which is a popular setting. This result is presumed to be because the parallel network branches are not on the same mapping, which is different from MixFeat.

The following methods mix two images. Sample pairing (Inoue, 2018) repeats two learning phases alternately, with one phase learning the average of two images and one of their labels and the other phase learning the input image and label as they are. Augmentation in feature space (DeVries & Taylor, 2017a) mixes two neighboring images in a pretrained feature space to expand the distribution of the feature maps. MixFeat is considered to be an extension of these methods that enables repeated mixing in several dynamic latent spaces.

Methods that mix two images and labels have recently been proposed. Mixup (Zhang et al., 2018a) and BCL (Tokozume et al., 2018) mix two training examples (an image and label) with random weights, and they are reported to reduce over-fitting appreciably. The manifold mixup method (Verma et al., 2018) mixes two training examples, as does the mixup method, but mixes feature maps randomly selected from some predetermined latent space instead of images. The difference between MixFeat and these methods is described in 2.2.

Manifold adversarial training (MAT) (Zhang et al., 2018b) learns the most sensitive adversarial examples in each feature map through two-step learning for each mini batch. In the first step, the most sensitive direction for each example in the given mini batch is found. In the second step, each

example is shifted in the most sensitive direction in each feature map. MAT has the same purpose as our method in that it makes a latent space that can distinguish each class, but the magnitude of the perturbation cannot be determined reasonably because the perturbation vector is determined from only the sensitive direction. Moreover, MAT increases the training time because of the two-step learning process.

# 5 VISUALIZATION

Finally, we visualize the feature distributions learned with the vanilla and MixFeat methods in Fig. 7. We trained a six-layer neural network with or without MixFeat on two-class toy data that have a two-dimensional checkerboard distribution. The vanilla network architecture is a three-fold stack of $\{\text{fc}(20) \to \tanh \to \text{fc}(2)\}$ while the MixFeat network architecture is a three-fold stack of $\{\text{fc}(20) \to \tanh \to \text{MixFeat} \to \text{fc}(2)\}$, where fc($k$) denotes the $k$-way fully connected layer. The figure shows that each class is distinguished from the others in each latent feature space obtained with MixFeat whereas, without MixFeat, each class is distinguished from the others only in the output spaces. We conjecture that this is why the classification performance improved with MixFeat.

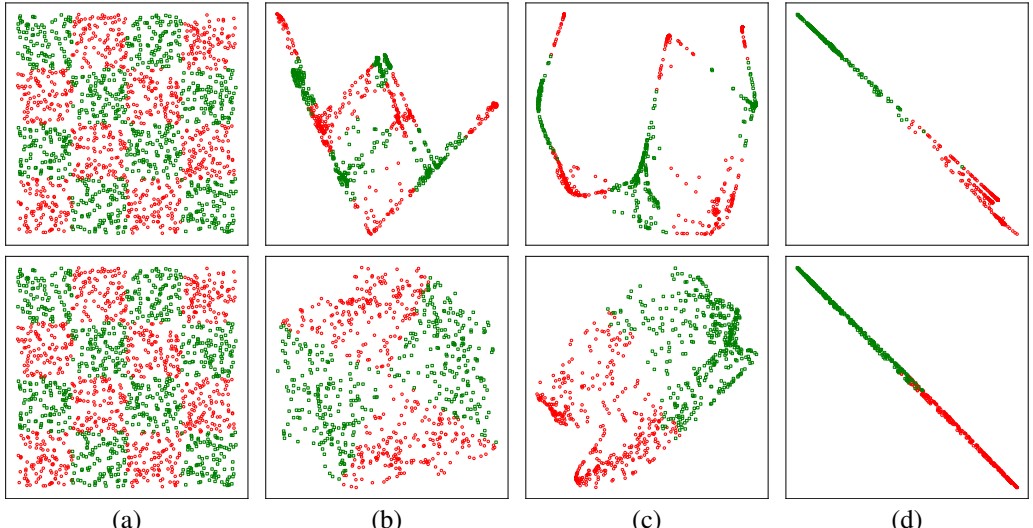

|           |           |           |           |
| :-------: | :-------: | :-------: | :-------: |
|    (a)    |    (b)    |    (c)    |    (d)    |

Figure 7: Visualization of feature distributions after 1,200-epoch training with a six-layer neural network on two-dimensional toy data. **Top row** shows the results without MixFeat, **Bottom row** shows the results with MixFeat. (a) The input distribution, (b) intermediate distributions after the second layers, (c) intermediate distributions after fourth layers, and (d) output distribution. In the distributions obtained with learning using MixFeat, it is easier to distinguish each class at each depth.

# 6 CONCLUSIONS

We proposed a novel method named MixFeat that mixes the feature maps in each latent space to avoid over-fitting in training deep neural networks. As a result, training the repeatedly mixed feature reasonably expands the feature distribution in each latent space and improves generalization performance. Our experimental results show that MixFeat appreciably improves the generalization performance. We discussed the relationship between our approach and a series of previously reported approaches and compared the MixFeat and mixup methods in detail. In future work, we are interested in verifying robustness to adversarial examples and verifying the problem of "manifold intrusion" as suggested in (Guo et al., 2018). Further studies are needed to extend MixFeat to tasks that use only small mini-batches, such as object detection or semantic segmentation. Because the MixFeat module can be easily added to various network models without additional computational cost in the inference phase, we believe that it will become the de facto standard for methods of reducing over-fitting.

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
