# OpenReview forum: "MixFeat: Mix Feature in Latent Space Learns Discriminative Space"
_ICLR.cc/2019/Conference_

### Official Review · AnonReviewer2 · 2018-10-31

**Rating:** 4
**Confidence:** 3

**Review:**

The paper proposes a method MixFeat for regularizing deep neural networks models, aiming at avoiding overfitting in training. The MixFeat interpolates, based on a careful selected mixing ratio, the hidden states (feature maps) of two randomly selected examples. Unlike MixUp, the MixFeat does not interpolate the labels of the two selected examples and the feature interpolation processes are conducted in the hidden space. Experiments on both Cifar10 and Cifar100 show that the networks with MixFeat improve their predictive accuracy as well as outperform networks with Mixup as regularizer.

The paper is well written and easy to follow, and the experimental results on both Cifar10 and Cifar100 show promising results. Nevertheless, the idea of interpolating pairs of latent features for network regularization is not very novel. Additional, the experimental section is a bit weak in its current form.

Main Remarks:

1.	MixFeat is very similar to Manifold-Mixup (Verma et al., Manifold Mixup: Learning Better Representations by Interpolating Hidden States), where both feature maps and labels of a pair of examples are mixed, so Manifold-Mixup would be a valid comparison baseline to MixFeat. In addition, the proposed method is similar to SMOTE (where features are mixed in the input space). In this sense, performance of SMOTE may be a useful comparison baseline as well.
2.	In the experimental section, the choice of parameter for Mixup seems arbitrary to me and may not be the optimal one. For example, for the Cifar10 and Cifar100 datasets, the original paper highlights that Alpha equals to one is a better choice to obtain better accuracy for ResNet. Also, as highlighted from AdaMixup (Guo et al., MixUp as Locally Linear Out-Of-Manifold Regularization), MixUp is quite sensitive to the choice of Alpha and suboptimal Alpha value easily leads to underfitting.
3.	Some claims are not well justified. For example, the authors claim that MixFeat can reduce overfitting even with datasets with small sample size, but did not provide any training cost or errors in Figure6 to support that claim.
4.	MixFeat is closely related to MixUp, and I would like to see more experiments with MixUp as baseline in terms of regularization effect. For example, it would be useful to include MixUp in Figures 4 and 6.

Minor remarks:

1.	What were the parameters for MixFeat used for Table 1?
2.	Is the proposed method robust to adversarial examples as shown in MixUp and ManiFold-Mixup?
3.	How the incorrect labels are generated in Section 3.2.1 is not very clear to me.
4.	Since MixFeat is similar to Mixup, I wonder if MixFeat has the problem of “manifold intrusion” as suggested in AdaMixUp when generating samples from image pairs?  How sensitive is MixFeat to the parameters Theta and Pi? Would learning mixing policies as suggested by AdaMixUp make sense here?

============after rebuttal============

I really appreciate the authors' rebuttal, which has addressed some of my concerns.
Nevertheless, I agree with the other reviewers about the main weakness of the paper. That is, why the proposed method works and what are its advantages over similar strategies, such as Mixup, AdaMixup and Manifold Mixup, are not clear.

---

> ### Author Response · Authors · 2018-11-26
> **Response to AnonReviewer2**
>
> Thanks for your helpful review.
>
> Main remarks:
> 1. First, we added the comparison result of manifold mixup on the CIFAR-10 and -100 datasets, and showed that our method MixFeat  performs better than manifold mixup. The reason why our MixFeat has better performance can be explained by the effect of repeatedly mixing feature as mentioned above.
> Second, we thanks for your referring to the SMOTE paper. However, the SMOTE algorithm targets the learning of imbalance data and that is, the SMOTE cannot be applied to the learning of the balanced dataset as it is. The SMOTE algorithm balance sizes of classes by adding new elements to small classes and by removing elements from large classes. So instead, we added comparison experiments with a special case of MixFeat that mix features only in the input space (in other words, the method mixing images and not mixing labels), named input-MixFeat. The results show original MixFeat performs better than input-MixFeat, by the effect of learning repeatedly mixing features in each latent space.
> We reflected them to section 3.1 and 3.2.
> 2. We are grateful for your pointing out mistakes about the parameter for mixup. As a result of reconfirming the original paper, as you pointed out, \alpha = 1 is a better choice of a parameter for CIFAR dataset. We fixed parameter of mixup and re-experimented, and the mixup performed better as in section 3.1 in revised version. As a result, there were cases where mixup had better performance than MixFeat in some settings, but in most cases MixFeat still had better performance.
> Please refer to table 1 for the summarized comparison results.
> 3. We guess you misunderstood our purpose of this experiment to show the possibility of reducing the number of datasets. But our purpose is to demonstrate *the effect of reducing the over-fitting problems* even when using small dataset. We apologize for a misleading section title of previous manuscripts, and we fixed the title of section 3.3.2  to prevent from misunderstanding, from “Reducing the size of the training dataset” to “With small training datasets”. When supporting this claim, you can understand that it is meaningless to compare training cost or errors with the full dataset.
> 4. Thanks for your recommendation to add the experimental results of mixup about regularization effect. We added the experimental results of mixup to training with incorrect labels (figure 4) and training with small dataset (figure 5 (left), which is figure 6 (left) in the previous manuscript and the order of figure has changed). Like MixFeat, mixup also confirmed the effect of suppressing over-fitting.
> We emphasize that MixFeat (perturbing features) and mixup (regression learning) are suppressing overfitting by different approaches as in section 2.2 in revised version. This fact suggests that combining MixFeat and mixup may further reduce over-fitting problems. We confirmed it by additional experiments as in section 3.3.
>
> Minor remarks)
> 1. We apologize for not clarifying the parameters used. MixFeat is placed directly after each convolution and \sigma=0.2 are used, that is the settings adjusted in the experiment of the latter sections 3.4.2 and 3.4.3.
> We reflected them to section 3.1.
> 2. Thanks for your helpful comment. Because MixFeat makes each latent space to easily distinguish classes, we believe the model learned with MixFeat is robust to adversarial examples as same as mixup family. Confirming this is left as a future work.
> 3. We apologize for not mentioning how we constructed the training dataset with incorrect labels. To construct a training dataset with a percentage p of incorrect labels from the original CIFAR-10 training dataset, we randomly selected a percentage p samples of training dataset and changed each label to another label that was randomly selected from the remaining nine classes.
> We reflected it to section 3.3.1.
> 4. Thanks for your helpful comments.
> Regarding AdaMixup, the problem of “manifold intrusion” is certainly wondered in MixFeat as same as mixup. However, we guess the problem can be reduced in MixFeat, by dispersing the degree of mixing in multiple layers. Of course, it is necessary that the verification for this hypothesis, which is left as a future work. Whether the mixing policies in AdaMixup makes sense depends on the result of the hypothesis.
> Regarding the sensitiveness to the parameters, we guess you have some misunderstanding about this. The distribution parameter in MixFeat is only \sigma, which is the magnitude of the distribution. We investigated the reasonable value of the \sigma in section 3.4.3. The symbol \theta means random angles obtained from uniform distribution U(-\pi, \pi). The symbol \pi indicates the circular constant. As this time, U(-\pi, \pi) has a uniform distribution over the entire angle.
> For the sake of clarity, we add the explanation of the \pi to section 2.3 and figure 2.
>
> Thanks for your other helpful comments. We had reflected them to the revised version.

---

### Official Review · AnonReviewer1 · 2018-11-02

**Rating:** 4
**Confidence:** 4

**Review:**

This paper proposes a method, so-called MixFeat that can mix features and labels. This method is in a similar line of the methods such as mixup and manifold mixup.

pros)
(+) The proposed method looks simple and would have low computation cost at inference phase.
(+) The experiment of evaluating the possibility of reducing the number of datasets looks good.

cons)
(-) The advantages of the proposed method are not clarified. There should be at least one insight why this method can outperform others.
(+) Decomposition of r and theta in eq.(1) looks interesting, but there is no supporting ground to grasp the implicit meaning of this idea. Why the parameters a and b are reparameterized with r and theta?
(-) Figure 1 is not clearly illustrated and confusing. Just looking at the figure, one can understand mixup is better than others.
(-) This paper does not contain any results validated on ImageNet dataset. This kind of method should show the effectiveness on a large scale dataset such as ImageNet dataset.

comments)
- It would be better to compare with Shake-type method (shake-drop (https://arxiv.org/pdf/1802.02375.pdf), shake-shake) and SwapOut (https://arxiv.org/pdf/1605.06465.pdf).
- The performance of PyramidNet in Table 1 looks different from the original one in the original paper (https://arxiv.org/pdf/1610.02915.pdf).

The paper proposes an interesting idea, but it does not provide any insights on why it works or why the authors did like this. Furthermore, the experiments need to contain the results on a large scale dataset, and from the formulation eq.(1), the proposed method looks similar to a single-path shake-drop or shake-shake, so the authors should compare with those methods.

---

> ### Author Response · Authors · 2018-11-26
> **Response to AnonReviewer1**
>
> Thanks for your helpful review.
>
> pros)
> - Regarding the experiment of the number of datasets:
> Thanks for your positive comments, however, our purpose of this experiment is not to show the possibility of reducing the number of datasets, but to demonstrate the effect of reducing the over-fitting problems even when using small dataset.
> We fixed the title of section 3.3.2  to prevent misunderstanding, from “Reducing the size of the training dataset” to “With small training dataset”.
>
> cons)
> - Regarding the insight why MixFeat can outperform others:
> First, both methods MixFeat and mixup reduce over-fitting by reducing feature distribution of each class.
> Second, MixFeat can further reduce feature distribution by repeatedly mixing features. Repeatedly mixing features in various layers obtains more diverse features than mixing features in one layer. Learning high diversity features makes latent space easier to distinguish classes and suppresses over-fitting problems more strongly. However, Tokozume et al. (2018) reported as follows: “the performance when we used only the mixtures of three different classes (N=3) was worse than that of N=2 despite the larger variation in training data”, which indicates that we cannot mix repeatedly in the mixup family of methods.
> We reflected them to section 2.2.
> Additionally, we emphasize that MixFeat (perturbing features) and mixup (regression learning) are suppressing overfitting by different approaches as in section 2.2 in revised version. This fact suggests that combining MixFeat and mixup may further reduce over-fitting problems. We confirmed it by additional experiments as in section 3.3.
> - Regarding the parameters a and b being reparameterized with r and theta:
> Motivation of the choice is to make the magnitude of the perturbation follow a normal distribution. It is natural to conform the distribution of random perturbations to a normal distribution, but there are two ways: conforming the magnitude of the perturbation to a normal distribution, or conforming the spatial distribution of the perturbation to a normal distribution. We compared these two ways and the former choice achieves better performance. Then, we adopt the present way to generate a and b.
> We reflected them to section 3.4.1.
> - Regarding that figure 1 is confusing:
> After reconsideration, figure 1 seems confusing. We replaced the figure to emphasize the effect shrinking the feature distributions of classes by learning repeatedly mixing features. The smaller feature distributions, the easier to distinguish classes. I believe the new figure helps your understanding.
> - Regarding ImageNet dataset:
> We apologize that experiments by using ImageNet were not included. We had tried to sign-up in official website http://image-net.org/download-images to download original dataset of ILSVRC2012, which is standard large dataset, but we could not receive a permission email. Instead, we download images based on the provided image URL as much as possible and conducted comparative experiments. We trained 152-layer ResNet with learning settings same as in He et al. (2016b). The validation error (Top-1/Top-5) on the partial ImageNet dataset were (23.9%/8.3%) without MixFeat and (23.0%/7.6%) with MixFeat, while the validation error on the original ImageNet dataset is (23.0%/6.7%) without MixFeat. It was confirmed that the MixFeat is effective even for large datasets. However, since this dataset is a subset of original ImageNet dataset, we did not reflect the results in our paper to prevent from confusing with original ImageNet results.
>
> comments)
> - Regarding comparison with shake-type methods:
> Thanks for your recommendation and we added the comparison results with shake-type methods on the CIFAR-10 and -100 datasets in section 3.2, and showed our MixFeat performs better than any shake-type methods. We guess shake-type methods need further epochs (for example 1,800 epochs mentioned in their original papers) than our experiments.
> - Regarding as the performance of PyramidNet:
> As described in section 3.1, the learning settings are different from original paper depending on the memory limitation of our GPU. Mainly, a batch size is set to 128 in original paper and 32 in our experiments. Although the learning rate was also reduced according to the batch size, this settings do not achieve the performances of original paper. On CIFAR-10 dataset, since MixFeat (2.92%) exceeds the performance of the original paper (3.31%) despite the poor baseline (3.64%), We are convinced that MixFeat can achieve further better performance with larger batch size of 128 using higher performance GPU.
>
> Thanks for your other helpful comments. We had reflected them to the revised version.

---

### Official Review · AnonReviewer3 · 2018-11-03
**Interesting idea, but missing clear explanations and important baselines.**

**Rating:** 6
**Confidence:** 4

**Review:**

This paper follows a recent trend to improve generalization by mixing data from training samples, in this case by mixing feature maps from different samples in the latent space. One of the feature maps is added as a kind of perturbation to the other one, so only the label from the main feature map is used as the learning target. MixFeat, the proposed method of adding ‘noise’ from another learning sample is tested on CIFAR-10 and CIFAR-100 with different architectures. The authors claim that the proposed method makes the latent space more discriminative. Multiple experiments show that it helps to avoid over-fitting.

The core idea of mixing the latent spaces of two data samples is interesting and the results seem to indicate that it improves generalization, but I have two main criticisms of this work. First, it is unclear as to why this this approach works (or why it works better than similar methods) and the explanations offered are not satisfactory. The phrase “making features discriminative in the latent space” is used repeatedly, but it is not obvious exactly what is meant by this. Design choices are also not clearly motivated, for example what is the advantage of defining a and b as was done? The second criticism is that comparisons to manifold mixup should have been included.

Approach:
- In “1 Introduction”, the second contribution of presenting “a guideline for judging whether labels should be mixed when mixing features for an individual purpose” is not clearly communicated.
- Figure 1 is a nice idea to illustrate the types of mixed feature distributions, but is not convincing as a toy example. A visualization of how mixed features are placed in the learned latent space for real data would be more informative. The examples showing 0.4A+0.6B and 0.6A+0.4B are confusing - it’s not clear exactly how it relates to the formulation in (1).
- In “2.3 Computation of MixFeat” there is no clear explanation on why the authors chose a and b. Can they just be some other random small values? Is it necessary to have this correlation (cos and sin) between two feature maps we want to mix? Questions like these are not clearly explained. Similar questions can be applied to formula (4) and (6).
+ Explicitly pointing out how backpropagation works for MixFeat in (2) (5) (7) and Figure 2 is helpful.

Experiments:
- The authors mentioned important related works in both “1 Introduction” and “4 Relationship with Previous Work”, but in Table 1, they compared the MixFeat with only standard Mixup. Manifold Mixup would be a  better comparison as it has better performance than standard mixup and is more closely related to MixFeat - MixFeat mixes features in every latent space while Manifold Mixup does in a randomly selected space (and standard mixup only mixes the inputs).
- The method could be described as "adding some noise along samples' latent feature directions". An interesting perspective, and would have been nice to see a comparison of MixFeat vs. perturbing with gaussian noise to see how much the direction towards other examples helps.
+ The experiments to demonstrate the effectiveness of MixFeat for avoiding over-fitting are strong (aside from the missing baseline). The experiments showing robustness to different incorrect label ratios and with different training data size are convincing.
- In Figure 6 center, the x-axis is  or ( for original MixFeat and 1D-MixFeat, and  for Inner-MixFeat), but the authors didn’t make a clear distinction in both the figure caption and “3.3.1 Dimensions and Direction of the Distribution”, having it wrong for Inner-MixFeat with “6.94% ( = 0.02)” which should be “( = 0.02)”.
+ The ablation study motivating the choice of where to apply MixFeat was appreciated.

Related works
+ Clearly presented and covered the relevant literature.
- It would be helpful if the differences between MixFeat and the Mixup family is more clearly stated.

---

> ### Author Response · Authors · 2018-11-26
> **Response to AnonReviewer3**
>
> Thanks for your helpful review.
>
> Main criticisms:
> - Regarding why MixFeat works better than similar methods:
> We admit that it was not enough in previous manuscript that the explanation of the reason why MixFeat works better than similar methods.
> First, both methods MixFeat and mixup reduce over-fitting by reducing feature distribution of each class.
> Second, MixFeat can further reduce feature distribution by repeatedly mixing features. Repeatedly mixing features in various layers obtains more diverse features than mixing features in one layer. Learning high diversity features makes latent space easier to distinguish classes and suppresses over-fitting problems more strongly. However, Tokozume et al. (2018) reported as follows: “the performance when we used only the mixtures of three different classes (N=3) was worse than that of N=2 despite the larger variation in training data”, which indicates that we cannot mix repeatedly in the mixup family of methods.
> We reflected them to section 2.2.
> Additionally, we emphasize that MixFeat (perturbing features) and mixup (regression learning) are suppressing overfitting by different approaches as in section 2.2 in revised version. This fact suggests that combining MixFeat and mixup may further reduce over-fitting problems. We confirmed it by additional experiments as in section 3.3.
> - Regarding the phrase “making features discriminative in the latent space”:
> After reconsideration, we removed those phrases in revised version because this phrase is unclear.
> - Regarding design choice about a and b:
> Motivation of the choice is to make the magnitude of the perturbation follow a normal distribution. It is natural to conform the distribution of random perturbations to a normal distribution, but there are two ways: conforming the magnitude of the perturbation to a normal distribution, or conforming the spatial distribution of the perturbation to a normal distribution. We compared these two ways and the former choice achieves better performance. Then, we adopt the present way to generate a and b.
> We reflected them to section 3.4.1.
> - Regarding comparison with manifold mixup:
> First, we added the comparison result of manifold mixup on the CIFAR-10 and -100 datasets, and showed that our method MixFeat  performs better than manifold mixup. The reason why our MixFeat has better performance can be explained by the effect of repeatedly mixing feature as mentioned in section 2.2.
> We reflected them to section 3.1.
>
> Approach:
> - Regarding Figure 1:
> We replaced the figure to emphasize the effect shrinking the feature distributions of classes by learning repeatedly mixing features. The smaller feature distributions, the easier to distinguish classes. I believe the new figure helps your understanding.
> - For other items already mentioned in “main criticisms”, please refer them.
>
> Experiments:
> - Regarding manifold mixup:
> It is already mentioned in “main criticisms”, so please refer that.
> - Regarding the direction of perturbation:
> Thanks for your interesting perspective. I added the comparison results of the perturbing with isotropic Gaussian noise to demonstrate the effect of directive perturbation along samples’ latent feature directions. As you and we expected, the method using isotropic perturbation by Gaussian noise performs worse than any method using directional perturbation by mixing features.
> We reflected the result to section 3.4.1.
> - Regarding Figure 6 center: Mathematical characters have disappeared in OpenReview format, but I guessed that is your pointing out to typo of \alpha to \sigma. Thanks for your indication, and we fixed the typo in figure 6 and section 3.4.1.
>
> Related works:
> - Regarding the differences between MixFeat and the mixup family:
> As already mentioned in “main criticisms”, the main advantage of MixFeat is gained by repeatedly mixing features that is difficult in mixup family. Please refer “main criticisms” for details.
> We reflected them to section 2.2 and we refer it in section 4.
>
> Thanks for your other helpful comments. We had reflected them to the revised version.

---

> > ### Comment · AnonReviewer3 · 2018-12-03
> > **Response to authors feeback**
> >
> > Thanks to the authors for their substantial efforts in revising the paper! I think it has improved, and as a result I have increased my rating by one point. However, there are still some open issues: MixFeat does not seem to show a strong improvement over original mixup in the revised results of Table 1, and as an anonymous reviewer pointed out, there is a discrepancy between the results reported here and the results in the Mixup and Manifold Mixup papers on the same dataset. Also, while the explanation of why MixFeat should outperform the other methods is a bit more clear now, it would be nice to see some experimental evidence to support this explanation.

---

> > > ### Author Response · Authors · 2018-12-04
> > > **Response to reviewer's feedback**
> > >
> > > Thank you for your feedback.
> > >
> > > > MixFeat does not seem to show a strong improvement over original mixup in the revised results of Table 1
> > >
> > > We believe that the improvement over original mixup is practically not important.
> > > First, MixFeat achieved a strong improvement from vanilla models.
> > > Second, we believe that the combination enhancement effect is practically more important than which method is better.
> > > That is, our MixFeat (perturbing features) and mixup (regression learning) are different approaches,
> > > and those two approaches can be combined and further reduce over-fitting problems as in section 3.3.
> > >
> > > > as an anonymous reviewer pointed out, there is a discrepancy between the results reported here and the results in the Mixup and Manifold Mixup papers on the same dataset
> > >
> > > The discrepancy is due to the difference in architecture bewteen PreActResNet18 in original mixup paper and  ResNet20 (pre-activation version) (He et al., 2016b).
> > > Briefly, PreActResNet18 has four times wider channels and 40 times larger parameters than ResNet20.
> > > Please see also the reply to the anonymous reviewer.

---

### Author Response · Authors · 2018-11-26
**Revised paper uploaded**

We have uploaded a revised version of the paper.

Major changes:
- Section 1: Removed second item of main contributions,  since we could not experimentally or theoretically support the effect of the guideline presented for judging whether labels should be mixed when mixing features for an individual purpose.
- Section 2.2: Added an explanation of the difficulty in learning when mixing   repeatedly features and  labels to differentiated MixFeat method from and mixup family methods.
- Figure 1: Replaced the figure to emphasize the effect of concentration of feature distribution by repeatedly mixing features. The caption was rewritten accordingly.
- Section 3:
 -- Fixed the parameter \alpha to 1.0 and re-experimented in accordance with the description of the original mixup paper.
 -- Added the description of the parameters of MixFeat used in the experiments.
 -- Added comparative experiments with various regularization methods: manifold mixup, shake-shake, ShakeDrop and swapout.
 -- Added comparative experiment with input MixFeat that mixes only input features in order to show the effect of repeatedly mixing features.
- Section 3.2: Fixed the title of this section to prevent misunderstanding.
- Section 3.3:
 -- Added the experimental results of mixup about regularization effects.
 -- Added the experimental results of the method combined MixFeat and mixup to confirm that MixFeat and mixup are different approaches to reduce over-fitting.
- Section 3.3.1: Added the description about how to generate a training dataset including the incorrect labels.
- Section 3.4.1:
Added comparative experiments with another distribution of perturbation i.i.d.-MixFeat, which is choose a and b from independent random Gaussian distributions to support the reason of choosing a and b.
Added comparative experiments with another distribution of perturbation that is isotropic Gaussian noise to support the superiority of directivity distribution.
Fixed typo about the parameter of inner-MixFeat, \sigma to \alpha.
- Total: 8 pages -> 10 pages (for main part)

---

### Public Comment · (anonymous) · 2018-12-03
**Concerns about comparison with baseline methods**

Author claim that their method works better than the baselines: Mixup and Manifold mixup on Cifar10 using a PreactResnet20 (Table 2). They report the test error of 7.02%  and  7.68% with Mixup and Manifold mixup.  MixFeat has test error 6.54 %. However, the test accuracies reported by the authors for Mixup and Manifold Mixup are much worse than the test accuracy reported in the Mixup paper (4.2 % with PreActResnet18) and Manifold Mixup paper (2.89% with PreactResnet18). So these results do not show the fair comparison of  MixFeat with Mixup and Manifold Mixup.

My suggestion is that, for the fair comparison,  the authors should try to reproduce the baselines in above-mentioned paper, especially since the code for both papers is available online.

One thing that could help to produce the baseline from above papers is to run the training for more epochs. This has been suggested in both Mixup and Manifold Mixup papers.

---

> ### Author Response · Authors · 2018-12-04
> **PreactResnet20 (0.27M params) and PreactResnet18 (11M params) are completely different architecture**
>
> Thank you four your comment.
> Our experiments were fair comparisons because we used PreactResnet20 as the base network that completely different architecture from PreactResnet18.
> PreactResnet20 is refered from He et al. (2016b) and composed of 3-stage CNN with 3-residual blocks in each stage, the sizes of feature maps are {32,16,8}, the numbers of filters are {16,32,64}, and the model-size is 0.27M.
> Mixup and Manifold Mixup refer K. Liu, (2017) (URL https://github.com/kuangliu/pytorch-cifar .) and composed of 4-stage CNN with 2-residual blocks in each stage, the sizes of feature maps are {32,16,8,4}, the numbers of filters are {64,128,256,512}, and the model-size is 11M.
>
> The code of PreactResnet18 :
> URL: hhttps://github.com/kuangliu/pytorch-cifar/blob/master/models/preact_resnet.py#L65

---

> > ### Public Comment · (anonymous) · 2018-12-06
> > **Further Suggestion**
> >
> > Thanks for your reply. I would suggest that in the interest of the research community, the results from the Mixup and Manifold Mixup papers should be added in this paper, clearly stating why those results are different from the results reported in this paper.

---

> > > ### Author Response · Authors · 2018-12-07
> > > **Response**
> > >
> > > Thank you for your suggestion.
> > > We will reflect it to the final version.

---

### Meta-Review · Area_Chair1 · 2018-12-13

**Confidence:** 4
**Recommendation:** Reject

**Metareview:**

The paper describes a method to improve generalization by mixing examples in the hidden space. Experiments on CIFAR-10 and CIFAR-100 showed that the proposed method improves the generalization of the networks. The reviewers found these results promising, but argue that the experimental section was too weak in its current form - notable lacking experiments on larger scale datasets such as Imagenet. Notably the paper should compare more with the relevant baselines to better understand its significance.